# Parental Access to Healthcare following Paediatric Surgery—The Precarious Role of Parents as Providers of Care in the Home

**DOI:** 10.3390/children10091578

**Published:** 2023-09-21

**Authors:** Stefan Nilsson, Mia Hylén, Inger Kristensson-Hallström, Gudrún Kristjánsdóttir, Pernilla Stenström, Runar Vilhjálmsson

**Affiliations:** 1Institute of Health and Care Sciences, Sahlgrenska Academy, University of Gothenburg, 405 30 Gothenburg, Sweden; 2University of Gothenburg Centre for Person-Centred Care (GPCC), Sahlgrenska Academy, University of Gothenburg, 405 30 Gothenburg, Sweden; 3Queen Silvia Children’s Hospital, Behandlingsvägen 7, 416 50 Gothenburg, Sweden; 4Department of Health Sciences, Faculty of Medicine, Lund University, 221 00 Lund, Sweden; mia.hylen@skane.se (M.H.); inger.kristensson_hallstrom@med.lu.se (I.K.-H.); runarv@hi.is (R.V.); 5Department of Intensive and Perioperative Care, Skåne University Hospital, 205 02 Malmö, Sweden; 6Faculty of Nursing and Midwifery, School of Health Sciences, University of Iceland, Eiriksgötu 34, IS-101 Reykjavik, Iceland; gkrist@hi.is; 7Department of Pediatric Surgery, Skåne University Hospital Lund, Lund University, Lasarettsgatan 48, 221 85 Lund, Sweden; pernilla.stenstrom@med.lu.se

**Keywords:** access, healthcare, paediatric care, parents, postoperative care

## Abstract

Access to healthcare can facilitate parents’ self-management of their children’s care. Healthcare access can be described as consisting of six dimensions: approachability, acceptability, affordability, availability, appropriateness, and aperture. The aim of this study was to analyse these dimensions of healthcare access experienced by parents caring for their children at home following paediatric surgery. The method-directed content analysis, conducted with the six-dimensional framework of access to healthcare as a guide, was used to analyse twenty-two interviews with parents of children treated with paediatric surgery. All six dimensions were represented in the results. Acceptability was the most frequent dimension, followed by appropriateness and approachability. Affordability, availability, and aperture were less represented. Although access to healthcare after paediatric surgery is generally appropriate and approachable, parents may experience insecurity in performing the self-management needed. Complementary forms of information provision, e.g., telemedicine, can be valuable in this regard.

## 1. Introduction

It has long been recognised that paediatric postoperative care is a team effort consisting of healthcare professionals collaborating with parents in caring for the child [1]. The child’s parents are an important part of the care, and their learned skills in care influence the health outcomes of their child. Therefore, healthcare professionals have a responsibility to give appropriate information and assistance in order to enable parents to better meet their child’s needs in postoperative care and to strengthen the needed partnership [2].

When a child is discharged from hospital, the parents’ responsibility for self-management increases. Family houses like the Ronald McDonald House can be an intermediate step where parents can prepare before the child’s discharge from hospital [3]. When the family returns home, parents frequently report a gap in support—they need continuous support from healthcare professionals in order to feel safe and find it difficult to take full responsibility for self-management [4].

Parents’ care of their children at home presupposes that parents are sufficiently informed by healthcare professionals. However, as the given information is often reported to be overwhelming and inconsistently presented, an improvement in the quality of self-management information has been suggested [5]. Although the purpose of the information is to support the parents’ use of their health literacy, a study from Greece found that half of the parents showed insufficient levels of health literacy in postoperative care [6]. Health literacy encompasses a wide-ranging concept that involves an individual’s capacity to access, comprehend, process, and apply health-related information [7]. The information provided in connection with discharge may not encompass all facets of health literacy. Nevertheless, this information could enable parents to employ their health literacy effectively in caring for their children. It has been associated with patient safety [8], and parents’ health literacy has been related to the child’s quality of care [9]. Therefore, to be knowledgeable and skilled managers of their child’s care at home and to promote favourable outcomes for the child, parents need to have access to assistance from healthcare professionals in addition to written information.

Access to healthcare includes the possibility of using services, as well as the level of actual utilisation. Factors such as income, insurance status, geographic location, or education level should be subordinate to the actual need for access to care in order for that access to be equitable [10]. Penchansky and Thomas [11] presented a taxonomic definition of access to healthcare in terms of the fit between the patient and the healthcare system. Based largely on this work, Levesque et al. [11] introduced five different dimensions of access to healthcare: approachability, acceptability, affordability, availability, and appropriateness. A sixth dimension, aperture, has recently been added as an important dimension in access to care, particularly as it is related to e-health. Aperture indicates an opening of pathways of communication in cyberspace, and this dimension demonstrates the uncertainty of not adequately comprehending how communication in cyberspace works [11,12].

Paediatric surgery is highly specialised medical care that is increasingly centralised nationally, implying longer distances between the home and the centre of expertise. In Sweden, centralised surgery is characterised by long distances due to a sparsely distributed population. This presents certain challenges for parents who may struggle to understand their child’s condition and how they can take responsibility as custodians and caretakers. Geographic distance to the site of paediatric care may also make it necessary for parents to explore new modes of communication with healthcare professionals. It is not always possible for the families to visit the hospital, and the follow-up after the surgery sometimes takes place by telephone or via the Internet.

Easy access to healthcare, despite the distance to highly specialised medical care, is important for safe and successful care at home. The aim of this study was to analyse the different dimensions of healthcare access as experienced by parents caring for their children at home following paediatric surgery.

## 2. Materials and Methods

### 2.1. Design

The study was qualitative, based on semi-structured interviews with parents who participated in a control group within the larger context of a controlled experimental clinical trial (ClinicalTrials.gov identifier: NCT04150120). ^4^ In the clinical trial, families of a child that had been operated on for a surgical diagnosis or malformation were either supplied with an e-health intervention in the form of an e-health device (intervention group) or received routine care with standard follow-up (control group) when being discharged from hospital. The semi-structured interviews with the latter group concerned parental experiences following the child’s discharge from hospital and were analysed within the six-dimensional theoretical framework of access to healthcare [11] (see Table 1).

### 2.2. Setting

Participants and their children were admitted to the national specialised medical centre at a university hospital between August 2019 and June 2020, i.e., partly during the COVID-19 pandemic in Sweden, restrictions being in place from March 2020. The Department of Paediatric Surgery is a tertiary centre for general paediatric surgery and a national centre for advanced gastrointestinal malformation surgery. Around 1200 operations on children are performed annually at the department, including around 150 advanced malformation procedures in neonates and premature infants. The department has a catchment area of 5 million residents living within 600 km. The families approached for this study were offered standard care adherent to diagnosis-based treatment programmes, including written and oral information about the diagnosis and procedure, contact information for the department with telephone numbers and mail addresses to specialised paediatric nurses at the outpatient department, as well as scheduled follow-up visits in person for specific conditions.

### 2.3. Data Collection

Parents were consecutively included from each clinical area and invited to participate. Inclusion criteria were individuals who were able to read and write Swedish and were the legal guardians (parents) of children below four years of age who were planned for discharge after paediatric surgery for anorectal, colorectal, abdominal wall, or urogenital conditions. Parents of children under the age of four were chosen as the care activities are mainly carried out by the adults, and the healthcare is based on the dialogue between hospital professionals and parents.

The semi-structured interviews were conducted by a researcher who had not been involved in the child’s care and was unknown to the parents. Each interview started with an open-ended general question about the overall experience of arriving home after leaving the hospital, followed by open-ended questions about parental experiences with health services. A total of 36 parents were invited to the study in two steps, i.e., (i) by e-mail, followed by (ii) a telephone call. This is a small patient population in Sweden, but the sample size was defined as suitable to answer the research questions with a qualitative method in this study. Interviews were conducted over the telephone between April and September 2020 and lasted 8–25 min (mean 13 min).

### 2.4. Data Analysis

To report the results, both quantitative and qualitative analyses were used. The meaning units for each dimension were quantitatively calculated and presented with descriptive statistics. A meaning unit embodies a pivotal sentence within the transcribed data, encapsulating the fundamental essence of the participants’ experiences related to one of the dimensions described in the framework. The data were analysed qualitatively with a directed content analysis [13]. The method was based on Hsieh and Shannon [13] and has then been further developed and tested for credibility in a previous study [12]. The six-dimensional framework [12] adapted from Levesque et al. [11] was used in order to evaluate accessibility to healthcare as expressed in the data material. In the first step of the analysis, each member of the analysis group did their analysis of the total coded material individually (SN, MH, and RV). In the next step, meaning units were divided into predetermined categories and calculated to map the representations of access for each dimension in the interviews. The results of the analysis were discussed within the analysis group until a consensus was reached. The approach decided on beforehand was that of majority rule, meaning that at least two of the authors in the analysis group needed to agree on a classification into a specific dimension before the classifications were presented for final discussion and confirmation by all authors.

In the concluding phase of our analysis, we meticulously computed all identified meaning units for each participant and the material in total. Statistical analyses were conducted using IBM SPSS version 27 (IBM Corp. Released 2019. Armonk, NY, USA: IBM Corp).

### 2.5. Ethical Considerations

Information regarding the study was given orally in a structured way and followed by written information before written informed consent.

## 3. Results

### 3.1. Family Characteristics

In total, twenty-two parents were interviewed, representing sixteen families, as both parents in six families participated in the interviews. The background characteristics of the families are presented in Table 2. The sixteen children had gastrointestinal or urological surgery: drainage of perineal abscess (×2), pyloric myectomy (×2), inguinal hernia repair (×2), urachus resection, laparoscopic gastrostomy, ileocaecal intussusception reposition, hemi-nephrectomy, nephrectomy (×2), hemicolectomy, Morgagni hernia repair, cystoscopy and urethra catheter placement, and a secondary abdominal wound closure after gastroschisis.

### 3.2. Dimensions of Access

In a total of 372 meaning units, all six dimensions of access to healthcare were identified when analysing the interviews. Furthermore, no additional dimension was detected. The distribution of meaning units (%) is presented in Figure 1. The presence of all dimensions in each interview was high; in 90% (n = 20) of the interviews, the presence of four to six dimensions was identified. In the rest of the interviews (n = 2), three dimensions were identified. The dimension of acceptability was the most frequent dimension identified, followed by appropriateness and approachability. Affordability, availability, and aperture were present to a lesser extent (Figure 1).

#### 3.2.1. Approachability

Approachability was indicated when the parents identified and presented their needs, i.e., they contacted the healthcare professionals themselves in order to receive support. The parents requested an early follow-up at home after discharge, already after a few days, as they would have liked to have had answers to issues raised after coming home from hospital. The parents suggested that an early follow-up meeting after discharge would be appropriate in that it would allow them to obtain an opportunity to ask relevant questions concerning their situation at home early on, which they could not foresee in hospital.

*would have probably wished that … given a follow-up relatively quickly after so that … or at least so that you could talk and ask a few questions about what you are worried about.* BKK3

The need for an early follow-up was sometimes reported to be due to a lack of knowledge, as well as uncertainty, about how to care for the child postoperatively at home. The parents, therefore, contacted the hospital soon after coming home for more information, as the information online and in the brochures was perceived as not being personalised or as unreliable. The available information did not consider their child’s specific postoperative needs, and the parents were afraid of missing important care needs. Additionally, parents did not fully trust the information they obtained from primary care regarding the surgical condition of their child.

One important issue referred to by the parents was the possibility of being able to reach the healthcare professionals at the hospital when needed. Some parents said that they had been given a telephone number directly to the ward, which enabled and allowed them to talk to someone who had exact knowledge about their child’s situation. The healthcare professionals’ knowledge of their child was a prerequisite for quick support in their time of need.

*I was, was sent around a little at first, a little back and forth, but then finally I got in touch with someone who had, who had been with, had something to do with, my son*. BKK5

#### 3.2.2. Acceptability

Acceptability indicated that parents accepted the care that was offered, as well as their own participation in the care of their child. Parents, in general, accepted their role as providers of care, and some expressed being used to taking care of their children from the time they had been waiting for the operation at home.

Parents described how desirable it was to come home and that they felt ‘done’ with the hospital—they could see that their child felt better. But parents also expressed that going home was a bit uncomfortable. They were very appreciative of the specialised care given at the hospital and grateful that, so far, their child’s recovery was going as expected.

*Well, it felt both really nice and really scary and really strange … well, mostly really good, but also a bit strange of course.* BKK11

Parents accepted the wait before being called back for check-ups, trusting that their healthcare providers had a plan and feeling that they were taken seriously when needed. In contrast, parents also reported needing more information regarding aftercare and experiencing a lack of any real plan. Waiting for a check-up appointment, therefore, seemed to give rise to some concern.

*Er, and, you did not get any real input about when, like the actual length of the care and treatment in recovery. I would have liked more information about that.* BKK97

Parents sometimes experienced a strong personal responsibility to resolve problematic situations, such as malfunctioning communication between different healthcare providers when the country’s medical chart systems were not synchronised. This increased their uncertainty and generated additional efforts and more healthcare contacts for the parents.

*Then, like… it becomes my task to search for the information although it is the same organisation… Like, it would have been better if the primary (paediatric) nurse could start off by saying ‘Yes, okay, where have you been hospitalised? Yes, you’ve been there. Yes, good, then I will contact them.’* BKK6

Although the recovery process at home went as planned most of the time, parents expressed feelings of uncertainty and concern regarding their ability to handle the care provider role. Parents worried about not being able to identify postoperative complications while being at home. Sometimes, parents were doubtful about going home with their child when they felt that issues such as wound care had not been resolved before discharge. Feelings of being rushed home were also expressed.

Parents who expressed high comfort and acceptance in coming home often had prior medical knowledge—some parents were registered nurses, assistant nurses, or physicians. As such, they expressed higher confidence in providing care to their child at home.

#### 3.2.3. Affordability

Affordability indicated that parents reported willingness, and also expected, to spend resources on health services. This included a balance between costs and benefits with regard to financial, social, and time-related efforts.

Parents in this study expressed no financial aspect of affordability—instead, their reports referred to their social and time-related costs. They expressed that having the responsibility of caring for their child was worthwhile, considering that they could be in their own home. It was important to be together: the two parents, the child who had been discharged from hospital, and his or her siblings, spending time as a family. Being ‘separated’ from the hospital felt good and made them focus on the recovery process, but they still felt there were no obstacles to contacting the healthcare if needed, and hence, being at home was affordable.

*… we chose to do it like this anyway, as we lived so near the hospital…very close… and we have another child at home, so we wanted to go home.* BKK35

When the parents found something not affordable, they figured out more feasible solutions, thereby resolving the situation and making it more affordable. For example, instead of driving long distances for check-ups at the hospital several times a week, they initiated other ways to communicate, such as sending digital pictures.

*… it was partly to avoid travelling three times a week [to the hospital], so we sent pictures instead.* BKK11

#### 3.2.4. Appropriateness

Appropriateness indicated that the health services initiated continued care following the discharge of a child from hospital and that the care offered met the parents’ perceived needs. The parents described both appropriate and inappropriate care. Parental reports of appropriate care referred to getting relevant and understandable information about the treatment that the child underwent in the hospital and about potential complications of the treatment and the child’s prognosis. Furthermore, parental reports of appropriate care involved getting contact information before discharge from hospital, including the name and telephone number of a person to contact.

Appropriate care was also reported as comprising follow-up by hospital professionals involved in the hospital treatment or operation or follow-up by ambulatory child healthcare services. Moreover, according to the parents, appropriate care included information about parental self-treatment of their child in the home, i.e., about how to care for their child, what to look for, and how to respond if something was unusual or a cause for concern.

*I thought that both nurses, doctors, and surgeons followed up well. We got information and what we should … like that we shouldn’t bathe him, but showering should be done and dabbing, so to speak, the wound there, and … yes, like that, what we should think about, and so on.* BKK16

On the other hand, inappropriate healthcare was also mentioned. This concerned getting divergent information, having sensitive information transmitted orally in a public area, receiving complex and important directions only orally and not in writing, or not being given information when needed. Inappropriate care also included receiving contradictory messages or recommendations when contacting different professionals.

*… one doctor says ‘Yes, but this is how it is and it should be done in this way. And we will do this and this and this’ … And then two days later you talk to another doctor, and then it is, and then it is not the same message that is being delivered.* BKK97

#### 3.2.5. Availability

Reports of available care included parents’ experiences of scheduled follow-up care at the hospital or ambulatory child healthcare services. It was frequently indicated by parents that the care at their home hospital—which they were encouraged to contact if living far away from the Department of Paediatric Surgery—was not as available when they needed further clinical observations, for instance, due to suspected or actual complications, delayed recovery, ambiguous signs, or other uncertainties, or when the parents needed assistance with tasks such as wound dressing.

*And then when we … changed the dressing on the wound ourselves at the Child Health Centre, or fixed the bandage at the Child Health Centre, and then we were even in XXX, and had … because he also had these bruises on his hips, so we were there once a week and like bathed him.* BKK6

Parents reported that their self-initiated contact requesting support was often due to not being able to adequately describe the child’s condition, either orally or with pictures. Therefore, they asked for a professional to observe the child in person.

*We were called for a follow-up visit to the person who performed the surgery. But besides that, we had no [hospital] contact that I remember, which I maybe thought was a little surprising.* BKK3

#### 3.2.6. Aperture

Aperture indicated an opening of pathways of communication in cyberspace, and the dimension also reflected the uncertainty of not fully understanding the workings and implications of such communication. Despite this uncertainty, the parents seemed to accept the electronic pathways available to them. They sometimes sent e-mails to healthcare professionals at the hospital with photos of their child’s postoperative status, such as pictures of wounds. They reported that they felt that the communication regarding these photos supported their care in the home and that they felt guided in their decision making by discussing the photos with the healthcare professionals. Although they used standard e-mail services for sending sensitive photos, such as images of their child naked, they usually did so without concerns about security risks.

*We always take pictures at ten o’clock in the kitchen. It’s never good to send pictures by e-mail. [laughs] I don’t think we can do that, [laughs] so but … and it’s … well, no, maybe it would have been better if … they said that maybe we could have had some kind of video call, that might be better.* BKK 11

## 4. Discussion

This study described parents’ experiences of access to healthcare following their young child’s paediatric surgery. Healthcare access was analysed through a six-dimensional theoretical framework [12], with acceptability being the most frequently reported dimension of access. Nevertheless, parents in our study sometimes reported access barriers when waiting too long for follow-up contacts or not getting continued and timely care from healthcare professionals at the hospital. These findings differ from the results of a previous access study by Hylén et al. [12] among parents of children discharged following hospital treatment and where an e-health solution was applied for communication with hospital staff. In the latter study, easy access to continued contact with healthcare professionals at the hospital was generally reported and appreciated.

The studied group included a few children with congenital disorders needing surgery. Previous research on children with congenital disorders and their families reports a need for enhanced communication, providing disease-relevant information, as well as self-management information [14]. A consequence of a lack of informational support could be that families visit the hospital or other healthcare providers more frequently. Then, several obstacles might arise, such as long travel times between the home and the hospital, something which is often not advantageous to the child and the family. Contact over the telephone is another option. Judging from the present results, reaching relevant healthcare professionals could constitute an obstacle since telephone times and personnel availability appeared limited. An alternative to both visits and telephone contact is e-health through telemedicine, which has proved a valuable alternative that saves both costs and time for families as well as for healthcare services [15]. Paediatric postoperative care provided via telemedicine has been reported to be associated with shorter waiting times and reduced parental absence from work without interference with outcome [16]. Telemedicine should be offered in a safe way, which has been pointed out in the literature and emphasised by our previous e-health research [12]. This is especially important when sending patient information using the Internet without encrypting solutions. To protect children and parents and improve patient safety, it is necessary to identify risks and security threats associated with telemedical options, especially before implementing digital solutions in clinical practice [17].

In the present study, the bridging from hospital to primary care was reported to be problematic due to the primary care’s limited information about the child’s operation and condition. Improvements and more specialised training for primary care professionals, as well as better access to hospital records, could contribute to better coordination and continuity of care for the children and parents involved. Another suggestion is the use of telemedicine or e-health to increase the transfer of knowledge about the specific child between the operating hospital and primary care or the local hospital, also allowing parents to be included [18].

The parents in our study looked forward to their child’s discharge from hospital, and they were willing to take on the responsibilities of caring for their child at home. Still, parents found themselves in a precarious situation at home, which they sometimes realised after a while when problems and uncertainties arose. Their precarious situation was indicated by their continued need for information and support during the post-hospital phase, by their experience of delayed contact and follow-up, or by not immediately being able to contact the desired professional(s). This raises the question of how well parents are prepared for their caregiving role at home following their child’s hospitalisation. Parents differ in their ability to obtain, process, and act upon health-related information, and in previous research, a large proportion of parents showed problematic and inadequate health literacy [19]. Added to that, parents’ other domestic and work-related responsibilities require skilful management, coordination, and cooperation [20]. This emphasises the importance of a careful assessment of each parent’s health literacy, preparedness, and situation by hospital professionals prior to the child’s discharge, as well as continuous integrated professional care and support following discharge, whether provided through the hospital or primary care services. These findings also underscore the value of practices of informing parents with an active communication core in order to build rapport with parents and across professionals [21].

The dimension of aperture pointed to uncertainty due to a limited understanding of how cyberspace works. Parents sometimes sent e-mails to healthcare professionals at the hospital with pictures of their child’s postoperative status, such as pictures of wounds, reporting that they experienced the communication regarding these pictures as very supportive in their self-care of their child at home and that it guided them in their decision making. However, they used standard e-mail services, usually without paying attention to the security risks of sending sensitive photos. It appeared that the high level of trust in the specialised healthcare professionals was translated into trust also in the pathways used for communication with them.

This can be compared to a previous study of access to healthcare as perceived by parents when supported by an e-health intervention in the form of an e-health device aimed at communication through chats, photos, and videos. That study indicated that parents who were supplied with an e-health device before their child’s discharge from hospital experienced safe communication pathways to the highly specialised department. Moreover, the healthcare professionals were reported to be supportive and easy to access for parents using an e-health device and telemedicine for communication [12]. The parents in the latter study also reported that they felt safer when they could communicate with the same healthcare professionals who had cared for their child at the hospital, a communication that was enabled and facilitated through e-health. Since parents knew the specialised healthcare professionals well, breaches in information transfer could be avoided [12].

The other five dimensions of access to healthcare in our study were also affected by the possibilities of communication with healthcare professionals. In the previous study by Hylén et al. [12], most communication was delivered by healthcare professionals via e-health. Compared to our study, the parents in that study felt to a greater extent that the information was acceptable, appropriate, accessible, affordable, and accessible.

### Strengths and Limitations

The interviews were conducted by an independent researcher who was not involved in the research project. It could be considered a strength that the researcher conducting the interviews had no former connections to the participants. Not being involved in the project, with no pre-understanding, facilitates a naïve perspective and new insights. Normally, the researcher conducting the interviews is somehow involved in the project, which could affect the interview situation [22].

The data were partly collected during the pandemic when apparent challenges were reported regarding both delays and access to healthcare, which might have influenced the findings. It is known that the pandemic rapidly increased the solutions for access to healthcare through e-health [23]. Still, the data were collected in a control group within a larger study, and no changes were reported concerning the follow-up procedures despite the pandemic. Furthermore, the barriers described by the parents in this study concerned the gap between hospitals and primary care, as well as a lack of preparations for their caregiving role at discharge from hospital. The probability of the pandemic affecting these areas could be regarded as low, and therefore, the findings should be stable despite the pandemic.

The composition of interviewees in each family was inconsistent, which is a limitation and may affect the research results. Another limitation is missing information from three parents about transport to hospital.

## 5. Conclusions

Parents of neonates and infants undergoing paediatric surgery appreciate going home following the discharge of their child from hospital, but they report a need for increased information about how to care for their child at home. More information would have increased their confidence in caring for their child, as they did not always perceive themselves as having sufficient skills to perform the self-management needed. Complementary forms of information provision, e.g., telemedicine, can be valuable in supporting parents in their self-management of their child. From that point of view, we recommend that e-health is more frequently used.

## Figures and Tables

**Figure 1 children-10-01578-f001:**
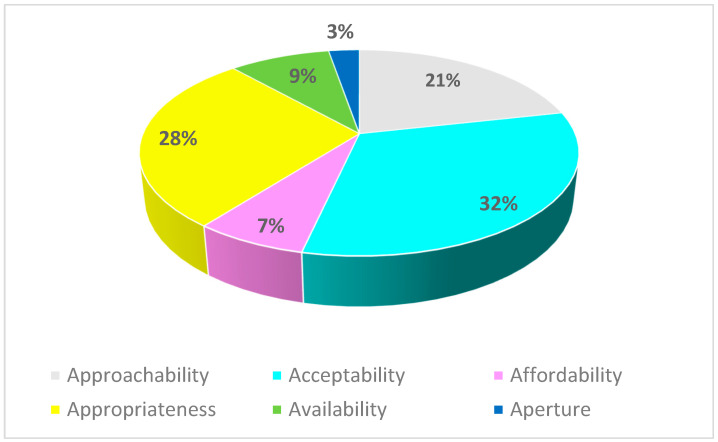
The distribution of access to healthcare presented for each dimension separately. Representation of the proportion for each dimension was calculated from all identified units in the interviews combined (%) (*n* = 22).

**Table 1 children-10-01578-t001:** The six-dimensional framework of access to healthcare [12]. This theoretical framework has been developed and presented by the authors in a previous study involving an intervention group. Therefore, this theoretical framework is presented in its original form in this study with the control group.

Approachability	Parents in need of healthcare can identify their needs and are able to, and do, initiate contact with health services that could have an impact on their child’s health.
Acceptability	Parents accept the care offered, as well as their own role as providers of healthcare, and where, how, and what services are provided.
Affordability	Parents have the time and resources to use health services and find the use of those services worth the cost and effort.
Appropriateness	The coordination and continuity of care initiated by the health services (through virtual and onsite contacts with known providers) generate a fit between the services provided and the parents’ perceived needs.
Availability	Healthcare is obtained and reached either in the home or at the healthcare facilities.
Aperture	The pathways by which communication is transmitted in cyberspace are not easily visualised by parents submitting information to health professionals and, therefore, create uncertainty.

**Table 2 children-10-01578-t002:** Characteristics of 19 of the 22 parents and all 16 children included in the study, and conditions of transportation to hospital (missing data from 3 parents).

Age (in years) and gender of the interviewed parents and their child	
Gender of the parents:	
- Female, n (%)	10 (53)
- Male, n (%)	9 (47)
Age of the parents, median (range years)	32 (24–40)
Age of the children, median (range months)	12 (0.25–48)
Transportation to hospital	
- Transportation time to specialised hospital, median (range minutes)	35 (15–180)
- Transportation by private car (%)	100

## Data Availability

The data that support the findings of this study are available from Lund University, but restrictions apply to the availability of these data, which were used under license for the current study and so are not publicly available. Data are, however, available from the “corresponding author” upon reasonable request and with the permission of Lund University.

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
