# Peer review of "Parental Access to Healthcare following Paediatric Surgery—The Precarious Role of Parents as Providers of Care in the Home"

_children, 2023, doi:10.3390/children10091578_

Round 1

Reviewer 1 Report

Thank you for the opportunity to review the manuscript entitled "Parental access to healthcare following paediatric surgery – the precarious role of parents as providers of care in the home" by Dr. Nilsson and colleagues.  In the manuscript the authors report the results of a qualitative study analyzing 6 dimensions of healthcare access experienced by parents of young children who had undergone pediatric surgical procedures.  The manuscript is well written and represents a thoughtful analysis of this topic.

Please address the following comments:

1) In the Introduction Section, the authors report that the goal of discharge instructions is to improve the parents' health literacy.  This argument/connection seems tangential.  Please consider revising this section.  To my understanding, health literacy is a broader concept of a person's ability to access, process, understand, and use health information.  Discharge instructions are specific instructions whose effectiveness may be influenced by the receiver's health literacy.  The purpose of discharge instructions is not to improve health literacy.

2)  Please define "meaning units"

Thank you for the opportunity to review this manuscript.

Author Response

Please address the following comments:

  • In the Introduction Section, the authors report that the goal of discharge instructions is to improve the parents' health literacy.  This argument/connection seems tangential.  Please consider revising this section. To my understanding, health literacy is a broader concept of a person's ability to access, process, understand, and use health information.  Discharge instructions are specific instructions whose effectiveness may be influenced by the receiver's health literacy. The purpose of discharge instructions is not to improve health literacy.

Thank you for your feedback. We have revised the text, aiming to provide a clearer explanation of the specific aspects of health literacy that we intend to describe: Health literacy encompasses a wide-ranging concept that involves an individual's capacity to access, comprehend, process, and apply health-related information. [7] The information provided in connection with discharge may not encompass all facets of health literacy. Nevertheless, this information could enable parents to employ their health literacy effectively in caring for their child.

  • Please define "meaning units"

Thank you for your feedback. We have defined meaning units: A meaning unit embodies a pivotal sentence within the transcribed data, encapsulating the fundamental essence of the participants' experiences related to one of the dimensions described in the framework.

Reviewer 2 Report

Your paper is very interesting and provides a novel description of parent caregiving needs after pediatric surgery. In particular I appreciate the use of your previously developed Six-dimensional framework of access to healthcare to characterize parent access to care for these potentially complex children in the post operative period. While the description of your 6-dimensional framework of access to health care is a unique and effective way to characterize parent responses to the interviews to answer your study aim, I think that the placement of Table 1 closer to Figure 1 or in fact combining them would assist the reader to interpret and draw important conclusions from your research findings. I had to flip back and forth between these two to interpret Figure 1. The shading does not distinguish well enough the 6 dimensions. Maybe use of patterns or linking the dimensions more closely to their label would help. I also really appreciated the use of the term aperture to suggest opening of pathways. This paper is a great example of a stepwise approach to studying a clinical problem. I look forward to future studies from your team in other clinical areas or from other researchers who will choose to use this interesting framework to study an important area of study, needed health care services for families of children with complex medical conditions. 

Minor grammatical errors: Page 2 of 11, line 71 delete the a. 

Table format using left justified would be easier for the reader and put the number labels next to the numbers in the parentheses. 

I very much appreciate how your paper builds upon your previous work. It is a great exemplar of a systematic approach to approaching an important research question. 

Author Response

Your paper is very interesting and provides a novel description of parent caregiving needs after pediatric surgery. In particular I appreciate the use of your previously developed Six-dimensional framework of access to healthcare to characterize parent access to care for these potentially complex children in the post operative period. While the description of your 6-dimensional framework of access to health care is a unique and effective way to characterize parent responses to the interviews to answer your study aim, I think that the placement of Table 1 closer to Figure 1 or in fact combining them would assist the reader to interpret and draw important conclusions from your research findings. I had to flip back and forth between these two to interpret Figure 1. The shading does not distinguish well enough the 6 dimensions. Maybe use of patterns or linking the dimensions more closely to their label would help. I also really appreciated the use of the term aperture to suggest opening of pathways. This paper is a great example of a stepwise approach to studying a clinical problem. I look forward to future studies from your team in other clinical areas or from other researchers who will choose to use this interesting framework to study an important area of study, needed health care services for families of children with complex medical conditions. 

We appreciate your feedback. We have relocated the table to Figure 1.

Minor grammatical errors: Page 2 of 11, line 71 delete the a. 

Thank you for your feedback. We have revised the text.

Table format using left justified would be easier for the reader and put the number labels next to the numbers in the parentheses. 

We appreciate your feedback. We have made changes to the format accordingly.